# SoK: Virtualization Classification on Isolation Capabilities

## Abstract

Within the Linux ecosystem, hypervisor and container-based virtualization are the two most prevalent and well-known server virtualization approaches. As it is often the case, the choice is much more complex than a binary decision between those distinct approaches. Recently emerging technologies, concepts and approaches, have greatly diversified the "server virtualization landscape". For example, the enabling concepts of container-based virtualization are ever changing and improve upon every upcoming Kernel release. Moreover, novel sandbox-based approaches leverage traditional and recent Operating System (OS) functionality to intercept system calls for their isolation needs. Hybrid systems utilize classic hypervisors in order to run a specific purpose built unikernel to run container-based virtualization within themselves.

In this work, we present an approach to classify virtualization aspects by their isolation capability. For this purpose, we decompose them into their respective enabling components and describe them in detail. Finally, we present a multi-level classification of server virtualization. This classification aims to enable a quick assessment of virtualization technologies and their induced implications.

## 1 Introduction

Virtualization technology isolation capabilities impose a challenge for many researchers, businesses and service providers alike. The isolation among processes, containers, Virtual Machines (VMs) or other containing units, is significant for a number of reasons. *(i)* Researchers aim for isolated experiments, without interference from unintentional foreign noise caused by other tenants. *(ii)* Businesses strive for the best possible infrastructure division while maintaining Service Level Agreements and maximizing profit. *(iii)* Service providers want to consolidate their infrastructure to keep total cost of ownership as low as possible. Naturally, poor isolation would negatively impact all the use cases above. These demands towards isolation are enabled by virtualization technologies.

Since its emergence in the 1960s, virtualization is ever-changing. Starting from experiments with time-sharing systems on mainframes [18], it has evolved into a broad landscape of technologies. These technologies are an integral part of the business models of many major organizations. Today, the application domains of virtualization are vast and the incentives for their adoptions are manifold.

This is particularly true for cloud computing and the direction it is progressing to. Areas and special research interest of the recent years include the Internet of things domain, fog computing, and edge computing. An encompassing term for these fields is the "Cloud-To-Thing continuum" [41]. Said tenants typically compete for resources for a variety of reasons like overbooking or arbitrary co-location [61]. Other emerging cloud computing models like Function-As-A-Service offerings also leverage virtualization to a great extent [50]. As Raza et al. further describe, they have complex demands for resource isolation, but also non-functional requirements like a fast cold start and low performance overhead. It is an essential requirement for virtualization software to be able to isolate them sufficiently.

Cloud computing and related domains are not the only fields where resource contention among tenants happens. In fact, distinct tenants on infrastructure are not necessarily distinct persons or customers. A simple but very common use-case is the demand to subdivide existing server hardware to improve its utilization [34]. For example, a company might operate a server with a database software, that is not able to fully utilize. This could be due to any reason like workload specifics or be imposed by the database software architecture itself. These underutilized resources could be used to operate another database software for another project, a scale out, or something completely different as a result of server consolidation [10, 13]. An incentive therefore could be better energy efficiency [38] and reduced total cost of ownership [34]. The trend towards the decomposition of monolithic applications and thus the enabling of distribution, as well as consolidation of application components, further diversified the virtualization landscape. These microservices pattern as described by Fowler and Lewis [27] are certainly widely applied in industry and research [57]. What is important though, is sufficient isolation among those applications, so that they do not negatively impact each other.

Besides the business oriented use cases, High Performance Computing (HPC) data centres and in consequence researchers utilizing them greatly benefit from the possibilities of virtualization. Every progress made in virtualization techniques is evaluated and frequently applied within these centres [29, 53, 65]. While they usually conclude, that native non-virtualized execution of experiments yields higher performance, this gap becomes smaller. In some cases the non-performance related aspects and the convenience of vir-

tualization can outplay the raw performance. Projects like Singularity [1] for example aim to provide reproducible environments for HPC experiments built upon virtualization features of the Linux kernel [37].

Even though all these application domains are highly relevant and represent a multitude of research areas, publication utilizing virtualization technologies often neglect the details of their respective implementations [39, 49]. Even within a seemingly narrow category like container based virtualization, implementations details make a huge difference regarding aspects like performance overhead and degree of isolation.

This paper follows a systematic approach in analysing virtualization technologies. We therefore review existing technologies and deconstruct them into their isolation enabling technologies. Along this perspective we aim to provide a multi-level classification of virtualization technologies. This classification enables an elaborated decision on which technology to choose and what to expect.

To provide a holistic view on the enabling aspects of virtualization technologies we make the following contributions to achieve a classification of those, based on their isolation capabilities:

- *Virtualization Technology Categorization:* We categorize virtualization technologies into three distinct categories: hypervisor-based, container-based and sandbox-based.

- *Elaboration on Virtualization Enablers:* For each virtualization category, we highlight the virtualization enabling aspects of those. These are integrated into the classification as subsidiaries.

- *Presentation of Dynamic Taxonomy:* Based on the categories and virtualization enablers, we present a multi-level taxonomy. We further introduce a cross-section hybrid-based approach that combines aspects of the previously established categories and thus integrates possible future developments.

The remainder of the paper is structured as follows: section 2 presents important background knowledge, frequently referred to in upcoming sections. This includes Linux fundamentals that describe essential levers for virtualization. Section 3 then presents a methodology on how the actual virtualization technology classification is pursued. This is followed by the implementation of said method in section 4. Based on this resulting classification, a brief overview over existing and widely adopted virtualization technologies is given in section 6. Within this section, said technologies are aligned to that classification followed by a short discussion. Afterwards, a review of related work is conducted in section 7. Finally, a conclusion is drawn in section 8 including a brief general discussion as well as some thoughts on possible future work.

## 2   Background

This section briefly presents some Linux OS specific fundamentals that tightly interact with virtualization concepts. We therefore highlight how the kernel interaction happens and how processes and memory are managed. Moreover, a short description of how the I/O devices disk and network are interfaced follows. All these resources are leveraged by virtualization approaches as described in the upcoming sections.

**Linux kernels**   are monolithic kernels. They manage Central Processing Unit (CPU) scheduling, memory, file systems, network protocols and system devices. Kernels are typically depicted as a layered ring graphs as shown in fig. 1. Notable here is, that applications are able to directly execute system calls or use an indirection via system libraries like `libc`[2].

System calls act as levers for applications to transit from user to kernel space. Further, the kernel provides an interface to the hardware, which in turn is interfaced via system calls again.

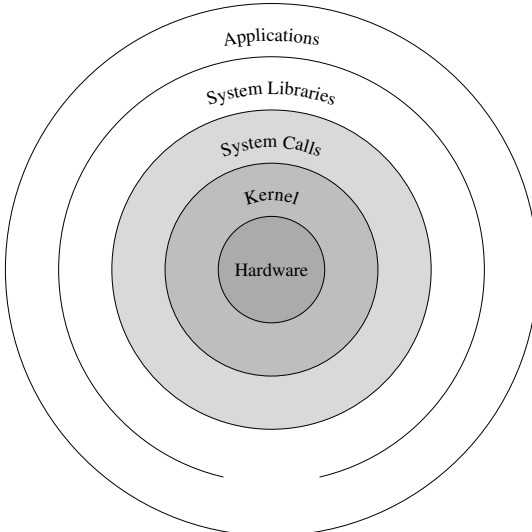

Figure 1: Linux Kernel

Based on this model there is a distinction made between *(i)* kernel mode and *(ii)* user mode. These special CPU modes provide distinct privileges to executed code. Executions within the *(i)* kernel mode are granted full access to devices and other privileged instructions, whereas user programs run in *(ii)* user mode. Execution in user mode runs unprivileged and needs to request privileges via system calls. Switching between user and kernel mode is called "mode switch". Examples for system calls include the opening of a file with `open`, mapping a file to memory with `mmap` or creating a new process with `fork`.

---

[1] https://sylabs.io/singularity/

[2] https://man7.org/linux/man-pages/man7/libc.7.html

**Processes** are the vessels for program code execution. Among other responsibilities, they manage address space, stacks and registers. Depending on the physical CPU attributes, processes can be executed in parallel, which is typically called "multitasking". They are identified by a unique Process Identifier (PID).

Processes can spawn other processes and threads. For Linux, all of these are resembled in the task data structure. All tasks on a Linux system together create a tree structure with the root PID being 1.

Thus, all tasks are created by other tasks using the system calls `fork(2)`[3] or `clone(2)`[4]. Internally, `fork` actually wraps `clone` with some privilege specific flags. After the creation of a new task with its own PID a system call like `execve(2)`[5]. This task creation flow is visualized in fig. 2. For the remainder of this paper, the term process will be used to refer to a running Linux task with a PID.

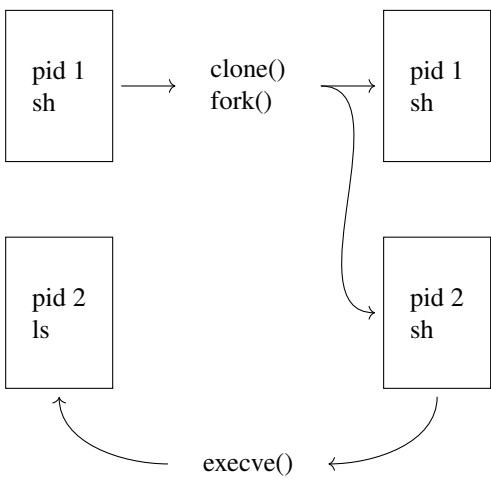

Figure 2: Task creation flow

**Memory** acts as a storage for kernel and application instructions. Alongside them resides their respective workload data. More specifically the "Main Memory" describes the actual physical memory of the system, commonly implemented as DRAM. It is segmented into "Pages" that typically represent 4 or 8 Kbytes, even though there are exceptions for "Huge Pages" if the CPU supports it.

Virtual memory on the contrary is an abstraction of the main memory and is presented as non-contended, almost infinite memory to processes. It is only mapped to physical memory on demand by the Memory Management Unit (MMU). Thus, virtual memory can be in four different states: *(i)* unallocated, *(ii)* allocated but not mapped yet, *(iii)* allocated and

mapped and *(iv)* allocated and mapped to a physical swap device.

Actually allocated and mapped memory is called "Resident Memory". The Resident Set Size (RSS) describes the total size of resident memory for a given process. This amount is of specific interest for isolation, since it is the actually contended memory resource.

The system call `mmap(2)`[6] is usually leveraged to allocate virtual memory. It is Linux' obligation when to map that allocated memory to the physical address space.

**Disk** or in particular disk I/O since it is attached to the I/O bus, represents the access to physical storage devices. The CPU is able to directly communicate with them via this bus. Within a computing system, they are typically represented as storage devices with an automatically generated name, following a system-specific scheme. Modern disks have a capacity in the GByte or TByte range and can be accessed by the kernel and applications.

I/O operations follow a standardized protocol and mostly consist of read and write commands. An I/O operation targets a sector which represents a small amount of storage on the physical device of typically 4 Kbytes. On top of a single or multiple disks, file-systems can be installed. They enable easy file-based, often tree like access to the disks.

Like disks, network devices are also attached to the I/O bus. Again, the CPU is able to directly communicate with them via this bus. The devices are usually referred to as Network Interface Cards (NICs). Within a computing system, they are typically represented as so-called interfaces or links with a name, generated by a system-specific scheme. The card itself, or the network controller, is defined by its transmission properties, or more specifically by its maximum possible throughput. Typical throughputs of models at the time of writing are 1Gbit/s to 100Gbit/s. Apart from that, NICs have one or more ports to connect to other NICs or a switching/routing device. Interconnections feature multiple connector interfaces like RJ-45 or SFP variations, as well as a transmission medium like copper or fibre.

Upon the intent of sending something to another link, the payload is split into packets of a previously agreed on size. In TCP/IP this is the "Maximum Transmission Unit (MTU)". These packages are further subdivided into nested frames depending on the applied network stack. For TCP/IP, this could be an "Ethernet Frame". These nested frames are then subsequently sent to a receiving NIC.

## 3 Methodology

In order to craft a representative and complete virtualization classification, a structured approach is necessary. Therefore, the method described lays out the steps that need to be taken.

---

[3] https://man7.org/linux/man-pages/man2/fork.2.html
[4] https://man7.org/linux/man-pages/man2/clone.2.html
[5] https://man7.org/linux/man-pages/man2/execve.2.html

[6] https://man7.org/linux/man-pages/man2/mmap.2.html

Foremost, a disambiguation of terms within the virtualization domain is important.

The term virtualization itself is rather broad and there is no general agreement on it across its applied domains. Many aspects of and resource types of computer systems can be virtualized. This ranges from the virtualization of full servers, over specific resources, towards certain aspects of applications. Within this paper, the focus lies clearly on the virtualization of servers or "server virtualization". While other aspects may be part of it, only technologies and approaches towards this goal will be considered. Here, server virtualization is defined as Ameen and Hamo [7] puts it:

> **Definition 1** (Server virtualization). *Server virtualization is the ability to run many operating systems with isolation and independences on other operating system.*

Based on this constraint, a comprehensive literature review is performed to lay out a possible server virtualization classification. This process starts with a very broad categorization and tries to narrow technologies down until sufficient distinction among them can be achieved. This criterion is met once the enabling technologies are identified.

The enabling technologies are investigated into detail in order to understand how they create isolation and what the implications are. These identified fundamental technologies act as a specific background and are presented as such in section 2

To begin with, the generally agreed on coarse classification of related literature, will be used as a baseline. It agrees on two distinct virtualization categories [20, 52, 54, 60], namely *(i)* Hypervisor-based and *(ii)* Container-based. These two categories and newly determined ones are further described during the remainder of section 4.

## 4 Virtualization Technology Classification

This section will investigate on possibilities to classify virtualization approaches. To begin with, it provides an overview that presents a quick glance at the resulting classification in section 4.1. Along this broad classification, each class is further analysed and investigated including their virtualization enabling components.

### 4.1 Overview

The anticipated classification is visualized in fig. 3. This classification acts as an overview and is derived from a broad literature research as described in the following sections. Precisely, the process to incrementally compose this figure is

described by stepping through these classes. Arbitrary starting from left to right, these are the three virtualization classes hypervisor, container and sandbox. Moreover, a fourth one named hybrid is part of this figure to indicate, that there are virtualization technology implementations, that share characteristics of all classes.

### 4.2 Hypervisor-based

Like virtualization in general, Hypervisor or Virtual Machine Monitor (VMM) systems have been around since the 1960s. During that time IBM had a huge impact on its development [19]. VMMs create an abstract layer between the hardware and nested OSs running on the same hardware. Resources of the host like CPU, memory, disk and network can be individually and dynamically attached to them. These OSs run with virtualized hardware and therefore instantiate "VMs". This term for hypervisor-based virtual servers is used from now on.

The following sections will briefly elaborate on various types of hypervisors, in order to distinct them. Further, a short discussion about how they achieve isolation follows. Within a short closing discussion, an initial iteration of the virtualization taxonomy is formed.

#### 4.2.1 Architecture Types

Goldberg, who was one of the most prominent researchers in the virtualization domain subdivided Hypervisor-based virtualization into two categories; Type-1 and Type-2 [28]. The main distinction among them is whether it runs directly on the hardware, or on top of another OS. Figure 4 illustrates that difference.

#### 4.2.2 Hardware abstraction levels

While these two distinctions categorize hypervisors, further significant properties can be found. Hwang et al. [33] describes some by highlighting three approaches on how the actual virtualization layer can be provided. These namely are *(i)* Full Virtualization, *(ii)* Paravirtualization and *(iii)* Hardware Assisted (HWA) Virtualization. These will be briefly discussed in the following.

*(i)* **Full virtualization** aims to run any OS and kernel, independent of its own physical system. No modifications to the guest system is necessary. With this approach, the host's and the guest's kernel and even their processor architecture can differ. This goal is achieved by binary translation and emulation, depending on its implementation [6, 7]. Hereby every device presented to the guest system is fully virtualized and created by the hypervisor. This for example includes CPU, mainboard, memory and NIC. If applicable, the translation between the virtual devices within the guest virtual system

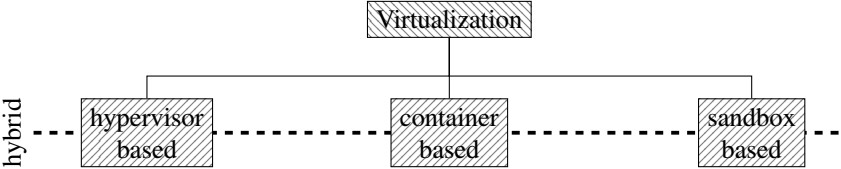

Figure 3: Virtualization Classification Overview

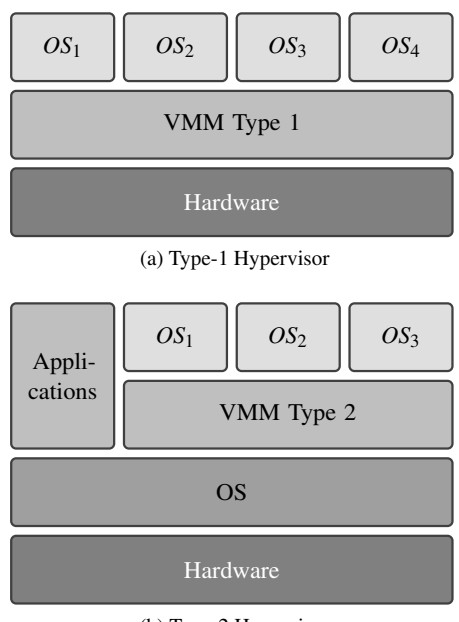

(a) Type-1 Hypervisor

(b) Type-2 Hypervisor

Figure 4: Hypervisor architectures

and the actual physical devices on the host system is done by the combination of guest and host drivers, managed by the hypervisor.

*(ii)* **Paravirtualization** aims to minimize the overhead virtualization of hardware brings [14]. It does so by providing and leveraging a special abstraction layer. This layer can be utilized by the VM to run privileged system calls on the hardware rather than in its own virtualized domain. These are also called "hypercalls". However, to achieve this, the guest OS has to be adapted and aware of these hypercalls. Depending on the implementation and configuration, the performance benefit can be significant [25].

*(iii)* **Hardware-assisted virtualization** is another way to reduce the performance impact on full virtualization. This technique came forward with the development of processor features dedicated to virtualization [15]. These features allow the trapping of certain calls without the need for binary translation or paravirtualization. While both, full- and paravirtualization can benefit from hardware-assisted virtualization, it can still be seen as a distinct category, since vendors decide whether to use that feature or implement it themselves within their hypervisor [62].

Neither of these approaches are mutually exclusive and specific implementations might apply different combinations or degrees of adaptation. However, these choices have significant impact on isolation characteristics as mentioned in section 5.4.

### 4.2.3 Classification Impact

To summarize, the following taxonomy for hypervisor based systems is crafted. However, while implementations that represent instances within this taxonomy share common isolation characteristics, specific implementation details impact the factual isolation. Figure 5 illustrates this taxonomy in a small tree like structure. Since hypervisor types and its means to provision virtualization are not mutually exclusive, every possible combination has to exist.

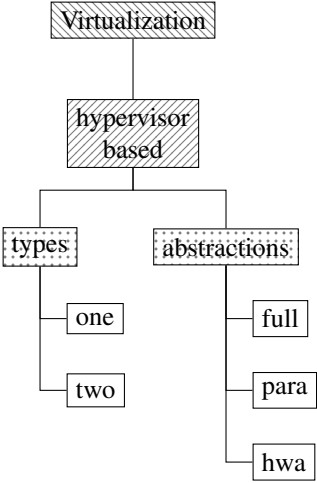

Figure 5: Hypervisor taxonomy

### 4.3 Container-based

Container, or more specifically within the context of this paper "Linux container" are isolated processes on a Linux

system that have their own view on most system resources. In contrast to VMs they do not utilize a hypervisor, but in consequence share the host kernel. However, since they are able to provide virtual servers including resource isolation, they are included within the virtualization taxonomy.

This section outlines distinct characteristics of container-based virtualization based on technologies applied. First, isolation targets and their relation to the technologies highlighted in section 2 are presented. Following up, the architecture of a typical container engine is discussed. Finally an extension of the previously mentioned taxonomy from section 4.2.1 is proposed.

### 4.3.1   Isolation targets

Compared to hypervisor based virtualization, container-based virtualization is fundamentally different. It does not allow for full virtualizations like the usage of any host kernel or different CPU architecture. It also does not make any use of paravirtualization, since no device or component is emulated. Furthermore, there is no hardware-assisted virtualization possible and necessary. Container-based virtualization solely makes use of the features of the host OS. However, the goals and use-cases for both approaches overlap to a certain degree. This namely is the provisioning of virtual servers [60]. Container based virtual servers are called "containers" from now on.

There are no virtual devices that are being presented to the virtual server as in hypervisor-based virtualization, since no emulation and binary translation is taking place. Instead, Linux kernel features are being used to limit access, view and utilization to the resources provided by and shared with the host. Dua et al. [23] present an overview of the aspects of resources that need to be handled by the kernel on an abstract level. More in-depth information can be found within this chapter in section 2. Specifically, these are *(i)* process, *(ii)* resource, *(iii)* network, *(iv)* filesystem, *(v)* storage, *(vi)* device and *(vii)* capabilities.

These aspects are briefly described in the following:

*(i)* **Process isolation**    creates a limited view of the process tree from the perspective of the container. All processes within the container are branched of a new process with the PID 1. This PID and its underlying tree is also visible from the host, but with different PIDs dependent on previous process state. This aspect is realized by using namespaces as described in section 5.1. More specifically, PID namespaces.

*(ii)* **Resource limitation**    affects all typically used resources of a server. This includes CPU shares, memory, disk I/O and net I/O. Access to those can be limited and isolated dependant of the applied virtualization technique. This aspect is realized by using cGroups as described in section 5.2.

*(iii)* **Network interfaces isolation**    is separate from the actual possible utilization of a device. The container needs its own personal network stack and only sees configuration affecting it directly. This aspect is realized by using namespaces as described in section 5.1. More specifically, network namespaces.

*(iv)* **Filesystem tree isolation**    provides containers with their own root filesystem to not interfere with the host. Files, installed packages and configurations of the host are invisible to the container client, if not explicitly configured differently. This enables the installation of packages and changing of configurations without interference. This aspect is realized by using mount namespaces as described in section 5.1.

*(v)* **Storage isolation**    gives containers their own storage area for any kind of state. This could be a mounted filesystem externally managed by the host. Apart from simple bind mounts, container engines frequently leverage more sophisticated storage engines, to provide containers with filesystems. These range from overlay filesystems promising maintainability benefits, while suffering from performance issues [45], to clustered ones like Ceph [68] where isolation is completely handled out of system. In simple cases this aspect is realized by using mount namespaces as described in section 5.1. More sophisticated approaches are directly offered by the container engine.

*(vi)* **Device isolation**    makes containers aware of specific devices on the host system. Specific ones like Intelligent Platform Management Interface (IPMI), Graphics Processing Units (GPU) or disks can be made available to the container. This aspect is realized by using namespaces as described in section 5.1. More specifically, mount namespaces.

*(vii)* **Capabilities**    describe which kind of operations the processes within the container are allowed to execute. These include operations like mounting a filesystem or binding to a network device. This aspect is realized by Linux capabilities as described in section 5.3.

### 4.3.2   Example architecture

Linux offers many levers to enact the actual isolation of all the aspects described above. Namespaces provide the necessary isolation mechanisms, cGroups regulate limits on resource utilization and Capabilities grant required permissions.

Figure 6 shows a superficial and slightly simplified container engine architecture on the example of Docker[7]. This architecture however can be easily adapted to other classic container engines that also make use of the three mechanisms

---

[7]https://www.docker.com/

mentioned above [5, 35]. The following will briefly discuss the elements of the Docker architecture.

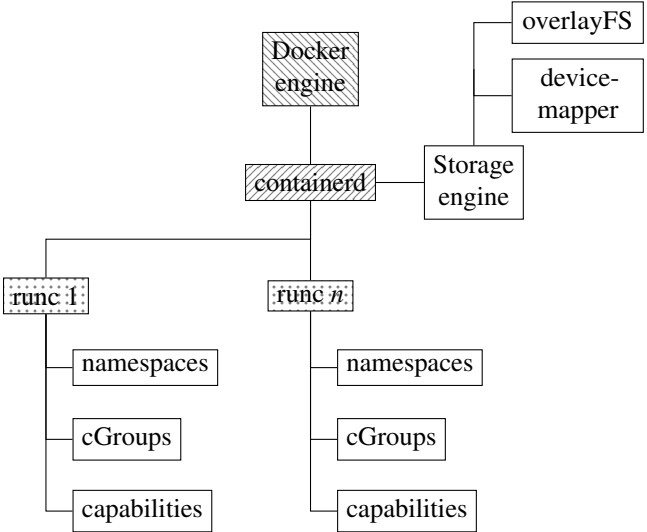

Figure 6: Docker architecture

The Docker engine itself is merely a Command Line Interface (CLI). Its primary purpose is user interaction and convenience of bringing all the container related features together. It therefore sensibly abstracts them to achieve an appropriate usability.

Containerd[8] is the actual daemon process running on a host, that is interacted with by using the Docker CLI. It therefore acts as a proxy towards the actual enactment of containers via runc and storage related features.

The storage engine enables containerd to provide storage for containers. This includes the authentication at container image registries to download base images a container is created from. Moreover, it provides access to storage for state, typically called volumes. Volumes could be provided using a local overlayFS or devicemapper concept, or be consumed from an external provider by leveraging specific storage drivers. Additionally, overlayFS is typically used to merge layered data including existing images, modifications and user data. This aspect is analysed by Mizusawa et al., who find many performance benefits in that approach compared to others existing at that time [46].

Finally, runc[9] is the component actually utilizing namespaces, cGroups and capabilities in order to create a running container.

Within the container domain, there are two important industry standards and specifications available. One being the *(i)* Container Runtime Interface (CRI), the other one being the *(ii)* Open Container Initiative (OCI) specifications. The *(i)* CRI defines an Application Programming Interface (API)

towards the container engine. In the example above, this would be containerd. This enables container orchestrators like Kubernetes[10] to transparently utilize different engines, as long as they are compliant to that API. The *(ii)* OCI on the other hand, describes how container images are supposed to look like, in order to be accessed and executed independent of the actual runtime like runc.

### 4.3.3 Classification Impact

To summarize, container-based virtualization fully depends on the degree Linux tools like namespaces, cGroups and capabilities are used. Moreover, storage is often handled outside the container perspective and is merely mounted into the respective namespace. This extends the taxonomy shown in fig. 5 as highlighted in fig. 7.

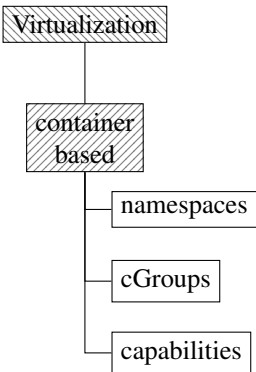

Figure 7: Container taxonomy

## 4.4 Sandbox-based

While most containerization technologies make use of the same Linux kernel fundamentals, there are some emerging technologies that pursue a different route. In order to better distinguish these from hypervisor and container-based virtualization they will be called sandbox-based from now on. However, this term is not an established term yet, but can be found among popular implementations of this approach.

### 4.4.1 Concept

Sandboxes can be created by utilizing system call filtering provided by the Kernel. Linux offers some mechanisms in order to do so. More background information on the system call filtering, and thus sandbox creation is presented in section 5.5.

These kinds of containers may still use all the principles highlighted in section 4.3 but are extended by the application of sandboxing methods. Wan et al. thoroughly investigate

---

[8]https://github.com/containerd/containerd
[9]https://github.com/opencontainers/runc

[10]https://kubernetes.io/

sandboxing possibilities for container for the purpose of isolation. They implement a two-step process. They first profile and record system calls a container executes, to limit those afterwards in a second step [66].

One representative technology of this class of container based virtualization is Googles gVisor[11]. Their approach is to reimplement fundamental Linux capabilities within the user space to gain more control and thus improve isolation [70].

### 4.4.2 Example Architecture

gVisor offers two operational modes. One is the `ptrace` mode discussed in this section. The other one utilizes Kernel Virtual Machine (KVM) in order to process system calls. This approach is discussed in section 4.5. A simplified architectural image is presented in fig. 8 as seen accordingly in its documentation [3]. As visible from that figure, there are two units between the application and the host; *(i)* Sentry and *(ii)* Gofer. These two and their relationship are briefly discussed in the following

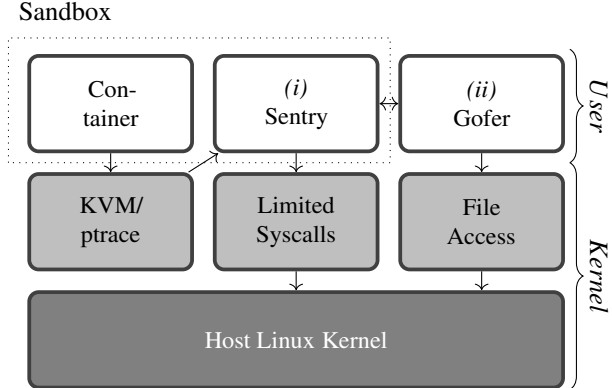

Figure 8: gVisor architecture

*(i)* **Sentry** itself implements Linux and is responsible for handling system calls. A container breaching security would only reach into Sentry and not into the host. It therefore exposes most of the Linux system calls, intercepts and reimplements them, in order to delegate them to the host.

*(ii)* **Gofer** is responsible for handling files outside of Sentry's own domain. Hence, it enables filesystem access for Sentry.

Due to the fact that many operations enacted by Sentry and Gofer are executed or proxied via user space, the performance overhead of such an approach is very high. Most

---

[11] https://gvisor.dev/

operations take at least double the amount of time compared to traditional container based virtualization approaches [70]. However, Young et al. also conclude, that sandboxes significantly improve security and isolation. Wang et al. [67] come to a similar conclusion.

### 4.4.3 Classification Impact

The sandbox based virtualization is a powerful method to improve isolation, but comes with a performance penalty. The approaches they use, most specifically the system call filtering, makes them an important addition within the virtualization classification and are thus added to the taxonomy. Hence, the taxonomy in fig. 7 is extended as presented in fig. 9

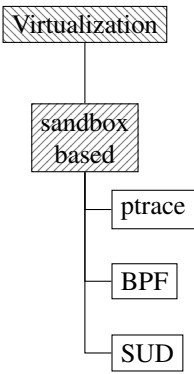

Figure 9: Sandbox taxonomy

## 4.5 Emerging and Hybrid Technologies

Besides the previously mentioned hypervisor, container and sandbox-based approaches, further technologies have recently emerged. Some of them claim to combine beneficial approaches of existing technologies, while minimizing their drawbacks. They often minimize choices in order to optimize and focus on details. On superficial observation however, they are not easily categorized among the previously introduced categories.

### 4.5.1 Concept

On deeper investigation though, all these solutions make use of previously existing technologies and thus follow the same approaches. As previously noted, these have impact on performance, security and isolation characteristics.

The combination of technologies enables the vendors to file opinionated decisions on specific implementations, yielding benefits for certain scenarios. Combining for example hypervisor and container-based solutions, the decision for a very specific OS within the virtual machine is made possible. The Kernel can be minimized to only enable necessities for container execution in order to reduce overhead and thus

combining isolation capabilities of both approaches. Kata[12] containers is a popular implementation pursuing that concept. While isolation capabilities improve, performance is slightly degraded compared to traditional container-based virtualization using runc as an example [36]. The previously discussed gVisor also offers a so called KVM mode, which follows a similar approach and is used as an alternative to ptrace.

A slightly more sophisticated form of the combination of existing technologies are unikernel or "library operating system" based systems. With the rise of cloud computing and convenient tools within this ecosystem they became a viable alternative to fully fledged Linux based VMs [42, 51]. Conceptually, they compile an application down to machine executable code being able to run directly on hardware without a general purpose OS involved. During that process only mandatory functionality is included. The resulting image can then be booted by a machine, which usually is a virtual one. The adaptation of virtual servers in this context is a key factor, since it significantly reduces the amount of hardware compatibility code necessary. However, as for the combination of virtualization approaches, this still relies on hypervisor-based virtualization and thus shares the same isolation capabilities. It does make a difference in the performance and security domain as shown by Compastié et al. [17] with their approach towards Software-Defined Security (SDSec). IBM's implementation called Nabla[13] represents a well-known representation of this approach.

### 4.5.2 Classification Impact

Even though not strictly being a virtualization class on its own, hybrid approaches shall also be included within the taxonomy. What is most important though, is the fact that any virtualization technology implementation may leverage any of the concepts highlighted within this taxonomy and described throughout this section. The resulting taxonomy is highlighted in fig. 10. Simultaneously, this figure also represents the final taxonomy and thus also includes hypervisor, container and sandbox-based virtualization.

## 4.6 Summary

This section proposes a taxonomy for virtualization technologies with respect to isolation capability. It therefore analyses existing approaches of prevalent technologies to categorize them as a first step. Those categories are *(i)* hypervisor, *(ii)* container and *(iii)* sandbox based ones. These are further subdivided into their enabling technologies and methods. Hence, the resulting taxonomy resembles a tree.

Within that tree, all leaf notes are considered to be options, whereas every other node represents a dimension.

However, modern solutions have evolved in ways, that utilize approaches of previously foreign domains. They do so in order to counter their own drawbacks or to optimize on different aspects. For this reason, a *(iv)* hybrid cross-section over all aspects as shown in the final taxonomy of fig. 10 is necessary. In consequence, the following definition for virtualization is proposed.

> **Definition 2.** *A virtualization technology's isolation is defined by the degree of realization of option leaves within the virtualization taxonomy dimensions.*

## 5    Virtualization Enablers

This section highlights the details for virtualization in relation to the virtualization classification of section 4. Hereby, we describe fundamental enabling technologies that are provided by the Linux kernel and leveraged by virtualization technologies.

## 5.1    Namespaces

Linux offers namespaces[14] in order to isolate system specific resources. It does so by wrapping them into an abstraction, in order to present them to a process [11]. This enables processes to yield completely different views of a system compared to the host system.

While this technology is an enabling one for containerization and thus Container-based virtualization, they do not directly relate. Both concepts and technologies can exist without the respective other one.

All available namespaces at the time of writing are highlighted in fig. 11. They are constantly adapted and extended in order to meet new demands and solve new challenges like a proposed CPU namespace [55].

The following paragraphs will briefly describe all those namespaces. The more prominently used and thus more important ones will be discussed into a little more detail.

*(i)* **cGroup**    namespaces[15] enable the usage of virtualized cGroups. When applied, a process is able to define its own cGroups, while the hosts cGroups are still active and protected. This allows for nesting of cGroups. For more information on cGroups in general refer to section 5.2.

---

[14]https://man7.org/linux/man-pages/man7/namespaces.7.html
[15]https://man7.org/linux/man-pages/man7/cgroup_namespaces.7.html

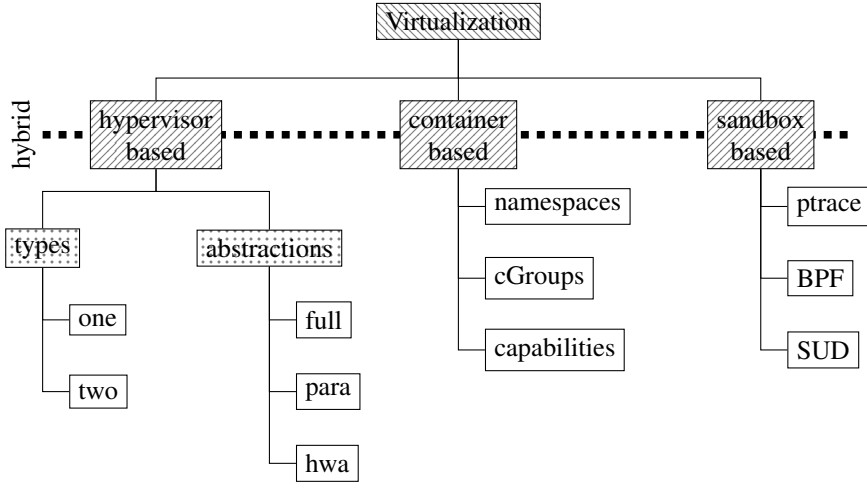

Figure 10: Virtualization Taxonomy

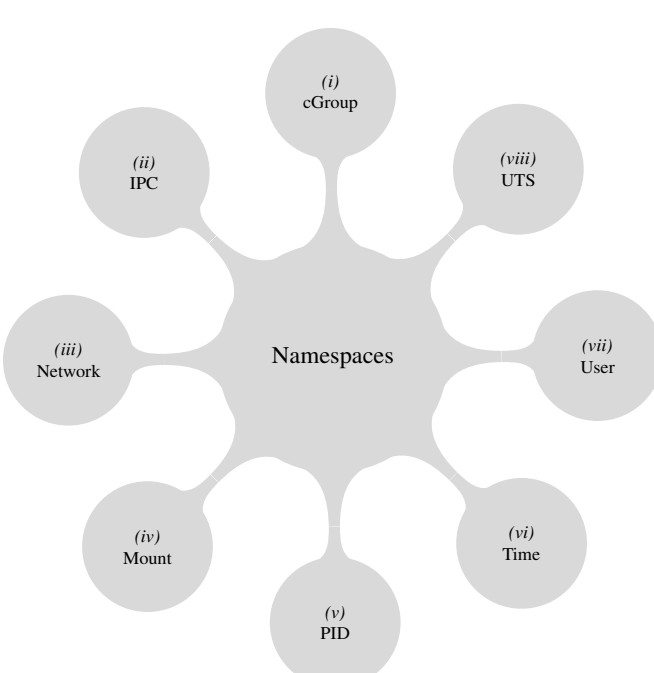

Figure 11: Linux Namespaces

*(ii)* **IPC** namespaces[16] isolate Inter Process Communication (IPC) resources. These mostly refer to message queues and the usage of shared memory between processes. By applying these namespaces, processes are able to generate their own identifiers for them without inheriting their parent ones.

*(iii)* **Network** namespaces[17] isolate networking related resources for a process. This includes interfaces, protocol stacks, routing tables and more. In practice, virtual veth[18] network interfaces are created, which pair physical or other virtual interfaces to form a pipe-like tunnel. This enables the creation of a bridge between those interfaces and in consequence, between network namespaces in order to create arbitrary virtual network topologies. Together with the mount, PID and user namespaces as described in the following, they provide essential levers for container virtualization.

*(iv)* **Mount** namespaces[19] isolate the list of mounts a process is able to see. Moreover, it allows the process to define its own mounts without interfering with other processes or the host. This important namespace allows to present a full root filesystem tree to a container including bind mounts for possible state as yet another layer.

*(v)* **PID** namespaces[20] isolate process related resources and abstractions. Processes in a PID namespace get their own PID starting at 1. Subsequently started processes invoked by

---

[16]https://man7.org/linux/man-pages/man7/ipc_namespaces.7.html
[17]https://man7.org/linux/man-pages/man7/network_namespaces.7.html
[18]https://man7.org/linux/man-pages/man4/veth.4.html
[19]https://man7.org/linux/man-pages/man7/mount_namespaces.7.html
[20]https://man7.org/linux/man-pages/man7/pid_namespaces.7.html

that process will have this new PID 1 as parent and will be assigned another unique one within that namespace. Collisions with other PID namespaces can not happen.

***(vi) Time*** namespaces[21] isolate the settings for the system clocks. This very recent addition to the Linux kernel mainline enables to set a process specific time which influences derived values like uptime. Moreover, it can also be leveraged for checkpoint restore methods for processes and container migration [44].

***(vii) User*** namespaces[22] isolate user related aspects for a process. These include user and group IDs, home directory, and capabilities. The latter is being described in section 5.3. This implies, that a user can have different capabilities within a user namespace than outside. In the case of a container, coupled with other namespaces, this allows an unprivileged host user to install packages within namespaces, that otherwise would require elevated privileges.

***(viii) UTS*** namespaces[23] isolates host and domain name. Processes within the same UNIX Time-Sharing (UTS) namespace are able to see and resolve to these names among them. Container engines typically leverage that to identify themselves. Moreover, container orchestration engines might use these namespaces to set up a cluster wide name resolution [43].

As already hinted throughout the description of namespaces, combining them makes them especially powerful. Using them in conjunction with cGroups extends that even more. This important building block for containers is discussed in the following section 5.2.

## 5.2 cGroups

Control[24] groups are a Linux feature that allows fine-grained control over different system resources [31]. More specifically, it enables to limit access to them. Typically, they are referred to as "cGroups". They are called "groups" because they can be applied to a group of processes which all share the same limits. Moreover, cGroups can be nested and are thus arranged in a hierarchical structure.

The cGroups project went under a significant restructuring effort, resulting in the release of cGroups v2. This effort was first merged into the kernel with version 4.5 and is able to fully replace v1 since kernel version 5.6 [22]. This paper

---

[21] https://man7.org/linux/man-pages/man7/time_namespaces.7.html
[22] https://man7.org/linux/man-pages/man7/user_namespaces.7.html
[23] https://man7.org/linux/man-pages/man7/uts_namespaces.7.html
[24] https://man7.org/linux/man-pages/man7/cgroups.7.html

focuses on the usage of v2 and thus this is the version being discussed in the following.

Resources are being controlled by resource controllers, sometimes also called subsystems. Figure 12 presents all those controllers visually. Like namespaces, they are constantly extended and improved like the most recent addition of a "misc" controller that is not yet part of most distributions [24].

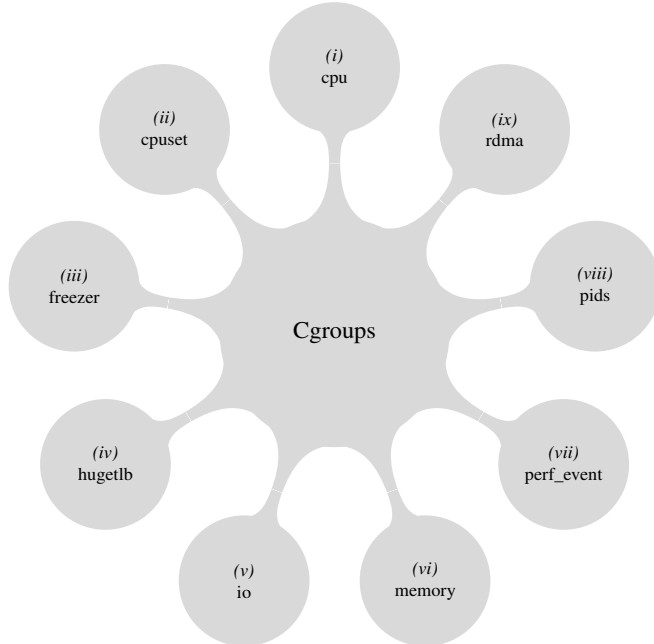

Figure 12: Linux Cgroups

The following paragraphs will briefly describe all those cGroups. The more prominently used and thus more important ones will be discussed into a little more detail.

***(i) cpu*** controllers set the amount of CPU cycles allowed. Apart from a raw value for cycles, aspects like weighted priorities and min/max utilization percentages can be set.

***(ii) cpuset*** controllers set constraints on CPU and memory placement. Only values specified are allowed for affected processes. This is especially helpful for Non-Uniform Memory Access (NUMA) systems [32].

***(iii) freezer*** controllers are able effectively freeze and thaw process groups. Oh et al. [47] have shown that this can be useful in order to dynamically increase system response time. They use freeze cGroups to freeze certain processes on demand to process a user input.

***(iv) hugetlb*** controllers limit size for huge pages for the affected group. This can have an effect on memory performance but is considered to be a complex topic. Panwar et al.

[48] elaborated on that and proposed a strategy to utilize them in and outside of virtualization. For a brief memory specific background refer to section 2.

*(v)* **io**   controllers enable the setting of both, bandwidth and Input Output Operations Per Second (IOPS) based limits to block devices for process groups.

*(vi)* **memory**   controllers set the amount of allocatable memory per process group. Moreover, it is possible to set hints for the Out Of Memory (OOM) killer. Without specific configuration, only processes within a cGroup are killed by it.

*(vii)* **perf_event**   controllers allow the gathering of cGroup specific perf events. These events are a means for kernel instrumentation and possibly contain sensible information like CPU counter or specific kernel function calls including payload.

*(viii)* **pids**   controllers are able to impose a limit on process generation for the affected process group. It can be configured with a maximum amount of possible `fork` and `clone` operations.

*(ix)* **rdma**   controllers regulate the access to Remote Direct Memory Access (RDMA) resources. This can be important for RDMA based devices like Infiniband NIC [40].

Similar to namespaces cGroups offer powerful measures to control, limit and possibly isolate resources. Used in conjunction with namespaces from the previous section 5.1, most enabling aspects for container virtualization are available. User specific capabilities are the last fundamental building block for isolation and are thus briefly presented in the following section.

## 5.3   Capabilities

Linux capabilities[25] are distinct units that allow the execution of very specific actions. These capabilities can be granted to a user or group.

At the time of writing the list of capabilities contain at least 80 different ones. They range from simple file operations over logging permissions towards complex admin like rights like kernel module loading. Certainly, this list is too extensive to discuss them here in a useful way.

Generally speaking, these capabilities exist to improve security. Fine-grained control over minimal operations allow system administrators to protect resources and to forbid certain actions. Hallyn and Morgan [30] have shown, that these

are very effective. Moreover, there is a strong synergy with user namespaces as described in section 5.1.

## 5.4   Hypervisor specific isolation

Isolation capabilities and the levers the hypervisor uses to achieve it highly depends on the choices described in the previous section 4.2.1. These do not always align with the possibilities Linux offers which may be due to arbitrary preference or the fact, that these possibilities have not been developed yet. Hence, the following will present some examples for the resources CPU, memory and I/O by highlighting how specific implementations solve isolation challenges.

**CPU:**   The Xen[26] hypervisor represents an interesting example for CPU isolation, since it offers the possibility to choose different CPU schedulers in order to control how this resource is shared. All approaches try to utilize this scheduler and therefore request shares. The scheduler will then schedule time for a VM for example based on its deadline, runtime or a credit system. Cherkasova et al. [16] discuss those schedulers in depth. They conclude, that the applied scheduler is highly dependent on the use-case but also state, that the default settings are not usable beyond experiments.

KVM[27] on the other side offers the possibility to utilize cGroups as described in section 5.2. This is possible due to the deep integration of KVM into the Linux kernel. Both approaches offer the possibility to dynamically adapt or change granted CPU shares to a VM.

**Memory:**   Silva et al. [59] state, that there are principally two distinct methods for memory isolation. One being cGoups as shown in section 5.2, the other one is static memory assignment. The hypervisor therefore requests memory from the host and completely blocks it for allocation to its managed VMs. Static allocation of memory is undesirable, since the risk of non-utilized memory is very high. Therefore, Waldspurger invented the technique "ballooning" [64]. They developed this for VMware ESX Server[28] a very popular enterprise VMM. By using their technique, they can prove that they can successfully reclaim or extend memory from a VM without negatively affecting it. The terminology they use for these respective terms are called "deflating" and "inflating".

**I/O:**   According to Waldspurger and Rosenblum two main approaches for I/O isolation can be pursued [63]. One being the *(i)* emulation of devices, the other one is *(ii)* paravirtualization. Both were previously discussed in section 4.2.1. Hence, limiting utilization can be provided by implementation details of the emulated device, or by leveraging cGroups.

---

[25]https://man7.org/linux/man-pages/man7/capabilities.7.html

[26]https://xenproject.org/
[27]https://www.linux-kvm.org/
[28]https://www.vmware.com/de/products/esxi-and-esx.html

## 5.5    Syscall Filtering

As described in section 2 system calls are used to create an interaction between user and kernel space. Linux offers the ability to intercept these system calls for debugging and manipulation purposes.

The latter allows them to be utilized for virtualization purposes similar to hypercalls as mentioned in section 4.2.1. This technique enables the creation of so-called "sandboxes", a mechanism applied in various application domains from embedded systems to cloud-computing [12]. According to Schrammel et al. there are three distinct levers available to intercept system calls. These are *(i)* ptrace, *(ii)* Seccomp-BPF and *(iii)* Syscall User Dispatch (SUD) [56].

*(i)* **ptrace**    is a system call itself[29]. It is able to examine other processes memory and registers, and is therefore primarily used for breakpoint debugging and system call tracing. Due to the fact that it can also filter system calls it is applicable to implement sandboxing.

*(ii)* **Seccomp-BPF**    is a kernel feature that allows for system call filtering. It therefore makes use of Berkeley Packet Filter (BPF) mechanisms. BPF or the recent implementation "Extended Berkeley Packet Filter (eBPF)", is a special VM running within the Linux kernel. This VM is able to execute code in kernel space that a user compiled in user space. This enables complex instrumentation and even runtime manipulation of kernel functionality. As a practical example, this technology is also used by modern browsers like the chromium project [2].

*(iii)* **SUD**    is the most recent addition to the Linux kernel [4] invented for Windows emulation. It enables the filtering of syscalls made from a specified memory region and can subsequently be dispatched to user space.

## 6    Validation

In order to validate the classification proposed in section 4 we present a list of popular virtualization technologies and arrange them along this classification. Table 1 highlights those in a tabular view. This list does not claim to be complete in any way but serves to purpose to get an idea of the existing landscape with respect to isolation techniques.

This table clearly reflects the variety of virtualization technology implementations and the fact that many solutions already implement hybrid approaches. However, distinct silos can still be perceived. Hypervisor and container based implementations tend to utilize isolation aspects from their own domain. There are exceptions though.

KVM[30] for example makes use of Cgroups if configured accordingly, while gVisor[31] utilizes a type-I hypervisor and sandboxing concepts. XEN[32] on the other hand acts as a traditional type-I hypervisor. Compared to KVM, it does not try to act as a general purpose OS. In this list VirtualBox[33] acts as an example for a widely adopted type-II hypervisor that is still capable of a wide range of virtualization techniques.

Docker[34], Podman[35] and Flatpak[36] are all representatives of container virtualization domain. Flatpak is slightly special here, as it aims to package graphical end user applications in contrast to the former ones. They do however utilize all the same Linux functionalities (Namespaces, Cgroups and Capabilities) to achieve isolation.

The sandbox domain is comparatively new in relation to the hypervisor and container based virtualization. gVisor[37] in particular is a very interesting representative here. It offers the possibility to utilize KVM as a type-I hypervisor to achieve sandbox functionality, but also presents the option to use ptrace instead. Both effectively perform system call filtering. While still being a research project and thus not being widely adopted, bpfContain [26] utilize modern BPF functionality to achieve this. Findlay et al. state, that they work on an integration into container runtime standards which seems very promising.

This list could certainly be extended indefinitely but gives an idea of the currently prevalent virtualization landscape

## 7    Related Work

Most publications arbitrary pick a list of popular or widely applied virtualization technologies in order to compare them. While they usually explain how the virtualization is enabled, these aspects typically come short [39, 49].

A similar situation applies to releases during and after the advent of container-based virtualization. Classifications of prevalent typically stop at a broader, more superficial point of view. The reason therefore is that they usually focus on something completely different instead of a mere classification of technologies [20, 52, 54, 60].

In contrast, Anjali et al. aim to classify virtualization technologies according to a scale based on "Location of functionality" [8]. Hereby they assume higher isolation, the less functionality is actually executed on the host kernel, compared to a guest Kernel. The scale itself ranges from low isolation like native Linux processes over gVisor hybrid approaches towards full KVM virtualization. While this claim

---

[29]https://man7.org/linux/man-pages/man2/ptrace.2.html

[30]https://www.linux-kvm.org/
[31]https://gvisor.dev/
[32]https://xenproject.org/
[33]https://www.virtualbox.org/
[34]https://www.docker.com/
[35]https://podman.io/
[36]https://flatpak.org/
[37]https://gvisor.dev/

| Name | Version | Comment | Hypervisor | | | | | Container | | | Sandbox | | |
|------|---------|---------|---|---|---|---|---|---|---|---|---|---|---|
| | | | I | II | Full | Para | HWA | Names-paces | Cgroups | Capa-bilities | ptrace | BPF | SUD |
| KVM | 2.3 | with Cgroups | x | | x | x | x | | x | | | | |
| XEN | 4.15 | | x | | x | x | x | | | | | | |
| VirtualBox | 6.1 | | | x | x | x | x | | | | | | |
| Docker | 20.10 | | | | | | | x | x | x | | | |
| Podman | 4.1 | | | | | | | x | x | x | | | |
| Flatpak | 1.14 | | | | | | | x | x | x | | | |
| gVisor | 2022 | with KVM | x | | | x | x | x | x | x | | | |
| gVisor | 2022 | with ptrace | | | | | | x | x | x | x | | |
| bpfContain[26] | 2021 | | | | | | | x | x | x | | x | |

Table 1: Virtualization technology classification of popular implementations

seems intuitive, they do not measure performance degradation impact by competing tenants, but rather performance overhead imposed by the technologies. Each of those technologies are highlighted by their own approach to achieve isolation, analysing the amount and call pattern of system calls. Combined with the results of this paper, their assumptions could be experimentally determined.

While this paper pursues a classification of virtualization technologies, the measurement of performance within virtualization technologies is tightly related. Various authors perform comparative studies regarding the performance degradation for virtualization technologies [9, 58]. Most find that containers are able to deliver almost bare-metal like performance, but also show promising results for hybrid solutions. Isolation on the other hand seems better for hypervisor-based virtualization. They do however imply, that there is a relation between the class of virtualization technology and performance. We, in contrast, classify virtualization technologies along their mechanisms for isolation, whereas they classify them along performance.

## 8   Conclusion

This paper aimed to craft a virtualization classification. It was done by dissecting established virtualization technologies and by studying scientific articles published in the virtualization domain. Implementing process led to a taxonomy that presents every substantial building block that enables isolation. On the most superficial level this taxonomy divided technologies into the three categories *(i)* hypervisor, *(ii)* container and *(iii)* sandbox based. Since applying enabling concepts within these categories are not limited to their respective category, *(iv)* hybrid based extends this list by one. The final resulting taxonomy is presented in fig. 10

Besides this summary, reflections on the resulting classification are discussed in section 8.1. Moreover, thoughts regarding possible future work are presented in section 8.2. This section takes on ideas that raised during the work on this

paper.

### 8.1   Discussion

What is yet to be shown though, is if this is also the case for any thinkable manifestation of virtualization technology. While this paper carefully crafted a classification taxonomy, the virtualization landscape is ever-changing. There might be minor adaptations necessary in order to assess any past or upcoming technology. Table 1 briefly shows a small proportion of those manifestations including upcoming ones.

Moreover, the classification performed here does not create an ordinal scale. An ordering based on isolation capability, startup time or performance overhead can only be performed based on measurements.

A central limitation regarding virtualization technologies certainly is the focus on the Linux OS. Other OSs also offer virtualization technologies including Microsoft's solutions like Hyper-V[38] or closed source Hypervisors like VMwares ESC[39]. Moreover, other UNIX based OSs offer solutions for container based virtualization like FreeBSD's Jails[1]. The methodology to measure those systems does not change, the profiling technology however needs to.

### 8.2   Future Work

Furthermore, since all the technologies presented in this paper are very Linux focussed, an adaptation to different OSs might be interesting. Especially technologies only applicable for Microsoft's OS Windows Server[40] could yield additional insights and possibly even a new class in the taxonomy.

As this taxonomy shows, there is a broad amount of virtualization technology manifestations possible. Especially considering different versions and configurations of those result in even more actually implemented solutions. This aspect

---

[38]https://docs.microsoft.com/virtualization
[39]https://www.vmware.com/de/products/esxi-and-esx.html
[40]https://www.microsoft.com/en-gb/windows-server

is an essential reason to not try to analyse every technology regarding their isolation capabilities, but rather craft a method to sensibly measure it on demand. Such an approach enables to compare isolation for a specific use case. This is not limited to isolation though. As mentioned in section 8.1 other ordinal scales like performance impact could be of interest. Even a "multi criteria decision making" approach could be applied like pursued by Domaschka et al. [21].

As mentioned before in section 8 Anjali et al. presented a scale for virtualization [8]. Combined with the findings of this paper and a sensible measurement methodology, both could be verified against their capability to isolate. Further, these could support the challenge of Williams et al. who questions virtualization for cloud computing in general [69]

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
