# OpenReview forum: "SoK: Virtualization Classification on Isolation Capabilities"
_JSYS/2022/Oct_Papers — Reject_

### Official Review · Reviewer_icES · 2022-11-02
**Interesting problem but needs a major improvement**

**Decision:**

Weak reject: interesting papers with flaws, not sure if they can be fixed in three months

**Review:**

## Summary

This paper presents an approach to classify virtualization techniques based on their isolation capabilities. The motivation is that recently it emerge many diversified virtualization technologies (e.g. hypervisor, container, and hybrid approaches), and it becomes hard for users to decide which approach to take. This paper provides a multi-level classification of server virtualization to enable a quick assessment of the technologies.

## Strength

- Choosing a virtualization technology is a valid and important problem that needs an SoK paper to provide knowledge for practitioners.
- This paper covers virtualization technologies across the spectrum: hypervisor, container, and sandbox.

## Weakness
- The problem is not well motivated & defined.
- It is unclear to me how this paper advances the field compared with previous work.
- It is unclear how this paper can achieve the claimed goal, namely helping practitioners decide which virtualization techniques to use.

## Detailed Review

Thanks for submitting to JSys! This is an interesting paper that targets an important problem: classifying the isolation capabilities of virtualization technologies. This paper also provides a broad-spectrum survey of techniques in hypervisor, container and sandbox. Here are a few questions the authors may consider to address to make the paper stronger:

### The problem is not well motivated & defined

The paper doesn’t state a concrete problem beyond a vague one: “complex choice”. From the abstract I got that the general problem is that “Recently emerging technologies, concepts and approaches, have greatly diversified the “server virtualization landscape”” and so “it is often the case, **the choice is much more complex** than a binary decision between those distinct approaches”. Then I was expecting a detailed discussion in the introduction about the complex choice, but I failed to find one.

* What factors do users need to consider when choosing between different virtualization technologies, like isolation, performance, or easy to deploy?

* Why does the paper choose isolation as the aspect?

The introduction only motivates why isolation is important but not motivate why a decision based on isolation is important for choosing a virtualization technology. There is a gap between them. Even within why isolation is important, it is still a bit vague to me.

* “Virtualization technology isolation capabilities impose a challenge for many researchers, businesses and service providers alike.” What is the challenge? Is it hard to implement or is it hard to choose a technology (the technologies are already there)? From the abstract, it seems to be the second case but this needs to be more clear (right now it’s more like motivating a paper that proposes a new isolation technique).

* “Naturally, poor isolation would negatively impact all the use cases above.” What does poor isolation stand for? Does that mean colocated applications may affect each other’s performance or one application can steal the secret from another one or both? It would be interesting to see a decomposed discussion of different aspects of isolation (e.g. performance, security) but I didn’t find one from the paper.

Many arguments in the introduction are not logically clear to me:

* “Areas and special research interest of the recent years include the Internet of things domain, fog computing, and edge computing. ” How are those directions related to virtualization?

* “As Raza et al. further describe, they have … but also non-functional requirements like a fast cold start and low performance overhead.” How are cold start and low performance overhead related to the paper’s topic, isolation?

* “Cloud computing and related domains are not the only fields where resource contention among tenants happens. In fact, distinct tenants on infrastructure are not necessarily distinct persons or customers.” I don’t really get the implication of this. Does that mean virtualization can happen not only at the tenant level but also at the application level?


### It is unclear to me how this paper advances the field compared with previous work

* “publication utilizing virtualization technologies often neglect the details of their respective implementations [39, 49]. Even within a seemingly narrow category like container based virtualization, implementations details make a huge difference regarding aspects like performance overhead and degree of isolation.” I don’t quite buy this argument. The referenced two papers provide very detailed evaluation of several specific virtualization techniques, like qemu, docker, gVisor. The results can directly help users to choose between the techniques. I’m not sure what “implementation detail” they are missing.

* “we make the following contributions…We categorize virtualization technologies into three distinct categories: hypervisor-based, container-based and sandbox-based.” I’m not sure this is the new contribution in this paper. The categorization is known before and is studied in existing work: Morabito, Roberto, Jimmy Kjällman, and Miika Komu. "Hypervisors vs. lightweight virtualization: a performance comparison." 2015 IEEE International Conference on cloud engineering.

* “For each virtualization category, we highlight the virtualization enabling aspects of those.” Right now this looks like a list of techniques used in hypervisor, container and sandbox. It’s unclear how these techniques are related to isolation and so unclear how useful the knowledge can be. I’d suggest the paper to have more in-depth discussion on how these techniques differs on isolation and what’s the tradeoff when choosing between them.

* “Classifications of prevalent typically stop at a broader, more superficial point of view. ” What does “broader, more superficial point of view” mean? Can the paper provide any example of that? And how does this paper solve the issue?

* “they do not measure performance degradation impact by competing tenants, but rather performance overhead imposed by the technologies.” How does this paper address the issue? I don’t see any performance measurement in this paper and I even seldom see a performance discussion in this paper.

### It is unclear how this paper’s knowledge can help practitioners decide which virtualization technique to use

The paper’s goal is to help practitioner choose a virtualization technique. I tried to pretend myself as a practitioner and see how I can do that based on the paper’s knowledge, but I found it’s hard to do that.

* The classification of hypervisor, container and sandbox seems to be known before; however, I still don’t know which one to choose after reading the paper. Which one has the best isolation? Which one has the best performance? What’s the drawback of each one?

* The taxonomies in each category (fig 5, 7, 9) look more like a list of techniques instead of a classification based on isolation.
  * How is type one and type hypervisors different in terms of isolation? I think they are the same in isolation since both virtualized the whole OS.
  * How would namespaces, cGroups, capabilities classify containers (Fig 7) ? I can’t choose one of them as a container, right? They are different implementation components of containers.
  * Same question exists in sandbox classification. How would I choose a sandbox based on ptrace, BPF and SUD (Fig 9)

* Table 1 has some interesting information that many virtualization products use multiple techniques cross hypervisor, container and sandbox. However, it is still unclear how I can make a decision based on the knowledge. Is it better to choose a product that uses fewer techniques or choose one that uses more techniques? I didn’t find an answer from either Section 6 or 8.

### Other Comments

P1 “High Performance Computing (HPC) data centres and in consequence researchers utilizing them greatly benefit from the possibilities of virtualization” grammar issue

The background section is a bit redundant. The concept of “Processes”, “Memory”, “Disk” are very basic concepts in computer science. The author may consider to discuss more on how they are virtualized instead of what they mean.

P4 “Within this paper, the focus lies clearly on the virtualization of servers or “server virtualization”.” I think this should be put in the introduction as the scope clarification. The whole scope of virtualization is large, and this sentence helps narrow down the scope. Before reading this, I was wondering where language-based virtual machines, like JVM, Python virtual machine fit in.

P4, P9 Why does the paper need to specifically list Definitions as Definition 1, 2?. Definition 1 is even purely a citation.

P11 Section 5.2 is verbose. I found it’s very close to the list on the linux manual: https://man7.org/linux/man-pages/man7/cgroups.7.html . I would suggest the author to describe this in a more concise way with a highlight of isolation.

P13 “In order to validate the classification proposed in section 4 we present a list of popular virtualization technologies and arrange them along this classification.” How does the validation work? Does a valid classification mean that popular virtualization technologies can be fit in the classification? How do you validate the classification can help differentiate isolation capabilities?


**Expertise:**

Follow the literature closely, last published 5+ years ago

**Useful:**

yes

---

### Official Review · Reviewer_YwtD · 2022-11-03
**Review for "SoK: Virtualization Classification on Isolation Capabilities"**

**Decision:**

Strong reject: this paper has serious problems, fixing it would definitely take more than three months

**Review:**

**Summary**

The paper classifies virtualization techniques in Linux based on the isolation guarantees provided by each technique. The paper covers hypervisors, containers, and sandboxing techniques.

**Strength**

- The paper attempts to classify virtualization techniques based on their isolation.

**Weaknesses**

- The SoK lacks breadth and depth.
- No novel insights.
- The writing is weak.

**Detailed comments**

I am not sure of the impact or contribution this paper makes. An SoK should contribute new insights by: qualitative analysis of existing research or empirical analysis of open-source software or identifying future research or challenges that require community attention or present a convincing, comprehensive taxonomy (https://www.jsys.org/type_SoK/)

Unfortunately, the paper fails on all counts.

The paper focuses on only Linux and only a few well-known virtualization techniques. The categorization into virtualization and containers is a well established one now.

The abstract mentioned that the paper's goal was to classify virtualization technologies based on isolation capability, whereas the introduction mentioned that the paper analyzes virtualization technologies and deconstruct them into their isolation enabling technologies. To me these sounds as different goals. The former seems to be about _what resources_ are isolated by each virtualization technology, whereas the latter seems to be about _how_ they enforce isolation. Regardless of the intent, the paper does not seem to satisfy either goals.

The paper discusses hypervisor and container based virtualization techniques, but the never manages to bridge the chasm between the two in terms of what resources are isolated in each and how. Then, clubbing the sandboxing approach with hypervisors and containers as a virtualization approach seems not very convincing.

On the other hand, there have been several advancements in intra-process or intra-address space isolation. How do these fit into the classification? As such, the paper lacks breadth and depth in its classification and taxonomy and provides no novel insights.

Moreover, it does not provide deep insights into any of the techniques covered, especially in the container and sandbox-based techniques. For hypervisors, the type-1 and type-2 architectures, and the full-, para-, and hardware-assisted virtualization are widely studied. For containers, much of the information provided is covered in literature and even blogs already.

Finally, there is clearly no empirical evaluation nor any discussion of future research directions in virtualization or isolation. The paper just cites some prior work on performance comparisons of virtualization technologies.

**Writing**

- The citation format is inconsistent. Some references do not have dates.
- Introduction: "Said tenants typically compete for resources for a variety of reasons like overbooking or arbitrary co-location." Tenants have not been mentioned before, so not clear what "said tenants" refers to. And tenants do not explicitly compete to overbook, so this sentence does not make much sense.
- "After the creation of a new task with its own PID a system call like execve(2)." -> this sentence is incomplete.
- Section 4.4: "However, this term is not an established term yet, but can be found among popular implementations of this approach." I don't know why authors think sandboxing is a popular term but not an established one.
- "One representative technology of this class of container based virtualization is Googles gVisor." No justification provided why authors think gVisor is representative of container technologies.

The authors should look into additional references for their study and elaborate on them:
- Nova Hypervisor: https://hypervisor.org/
- Firecracker: https://www.usenix.org/system/files/nsdi20-paper-agache.pdf


**Expertise:**

Follow the literature closely, last published 5+ years ago

**Useful:**

no

---

### Official Review · Reviewer_jR75 · 2022-11-12
**Lack of technical depth and organized writing for virtualization taxonomy**

**Decision:**

Strong reject: this paper has serious problems, fixing it would definitely take more than three months

**Review:**

Thank you for submitting the paper to JSys.

### Summary of the paper
The paper classifies Linux server virtualization technologies and conducts a taxonomy of hypervisor-based, container-based, and sandbox-based virtualization technologies. It also lists different resource isolations associated with each virtualization category.

### Strengths
Virtualization is essential in Linux systems. The paper has a detailed categorization for server virtualizations.

### Weaknesses
**[Lack of technical depth]** The paper has abundant but on-the-surface materials. I appreciate that the authors collect detailed virtualization and isolation techniques. However, piling up different technologies and terminologies makes the paper cluttered. I would suggest the authors dig deeper to find the connections and relevance of virtualization categories, especially their intersections and modern use cases. Much background information can be reduced or removed. For example, Section 2 can be removed while Section 3 can be shortened into one paragraph. The paper would excel if the authors could extend each virtualization category to include their modern use cases, e.g., kata container, unikernel, and to further conduct experiments to compare those technologies.

**[Unorganized writing]** The overall writing is scratchy. From Introduction to Future Work, every section even every paragraph has grammar issues. Moreover,  I am not sure whether all 12 figures in the paper serve any purpose. They are disconnected, redundant, and very spacy. Please also reduce the 37 footnotes. Throughout the paper, the authors fail to make virtualization and isolation a cohesive theme, but discrete and shattered (sub)sections.


**Expertise:**

Actively publishing in this area

**Useful:**

yes

---

### Official Review · Reviewer_exqD · 2022-11-13
**Review: Virtualization Classification on Isolation Capabilities**

**Decision:**

Weak accept: good paper with flaws that can be fixed in three months

**Review:**

### Paper summary
Virtualization technologies are widely applied to and researched in different areas, including cloud computing, server utilization improvement, and high performance computing. Despite the application and research effort, there lacks a systematic approach that analyzes the virtualization technologies in detail, in particular, how isolation is achieved. This paper presents a systematic analysis on different virtualization technologies and an anatomy on their virtualization enablers, creating a classification for virtualization technologies. It then discusses the enablers in greater details and uses real-world virtualization implementations to validate the classification.

### Strengths
- The paper presents a systematic top-down approach that dissects the virtualization technologies into their corresponding virtualization enablers.
- The virtualization enablers are presented in a detailed and clear way. Though I am already familiar with virtualization technologies, I still learned something new from the paper
- I found the discussion about isolation strength in section 8.2 interesting, as this is one direction where the analysis is useful for.

### Weakness:
- The use cases / implications of the analysis are still a little weak to me. Certainly the last paragraph of section 8.2 makes a fair point, but I believe the use cases and implications can be elaborated in greater detail.
- Though the paper did an good job on discussion of the virtualization enablers, the syscall filtering section (section 5.5) is not accurate / missing details:
    - Ptrace discussion should discuss the performance overhead, given that trapping to the tracer requires context switches between userspace and kernel space.
    - Seccomp discussion is not accurate. Seccomp currently (i.e. in the mainline kernel) uses the old classic BPF (cBPF) for system call filtering. There have been efforts to use the newer eBPF for system call filtering. The performance benefit of Seccomp-BPF is also missing.
    - As far as I know, both Docker (runc) and Podman (crun) supports Seccomp-BPF syscall filtering by early 2021, but this is not marked in Table 1.

### Minor problems (more on the paper writing side):
- The virtualization enabler is not clearly defined before section 4. Although it becomes clear to me in later sections, I suggest the authors provide a formal definition in section 3.
- Definition 1 seems detached from the discussion on containers and sandboxes – there are no “multiple operating systems” for them.
- The virtualization enablers (i.e. ptrace, BPF, and SUD) in figure 9 are not discussed in section 4.4.
- The discussion of capabilities (section 5.3) is a little too thin compared to that of cgroups and namespaces. It would be better to bring up some of the important capabilities.


**Expertise:**

Actively publishing in this area

**Useful:**

yes

---

### Meta-Review · Area_Chair_pySj · 2022-11-14

**Recommendation:** Reject
**Confidence:** 5

**Metareview:**

**Meta-review Security area chair:**

The paper presents a systematization of virtualization techniques based on isolation capabilities, focusing on the Linux kernel. The paper targets isolation by decomposing and describing reltaed components in detail.

While the paper does a great job at introducing the components and explaining them, a systematization should go beyond "describing" and "surveying" existing systems but must introduce an evaluation framework along with a higher level abstraction. The reviewers felt that these aspects were missing in the current form.

Focusing only on container versus hypervisor neglegts all in-process virtualization approaches or trusted execution based approaches. This seems like an artificial limitation of the current paper that should not be necessary.

Given the discussion of the reviewers along with the reviews, this paper should not be accepted in the current form. To improve, the authors should broaden their review (including alternate containers) along with providing an actual systematization (and not just a survey/background description). The current paper serves as a great background section and we encourage the authors to work on the systematization.

---

### Meta-Review · Area_Chair_GSZU · 2022-11-14

**Recommendation:** Reject
**Confidence:** 5

**Metareview:**

**Meta-review Configuration Management area chair:**

The SoK paper classifies virtualization technologies by their isolation capability. The article first decomposes the existing technologies into their constituting components. It also presents a multi-level classification of server virtualization. The paper's central claim is that the classification enables a quick assessment of virtualization technologies.

The reviewers believe the topic is important and still relevant to the community. They liked that the manuscript covers virtualization technologies across the spectrum, including hypervisor, container, and sandbox. In addition, the focus on isolation was appreciated by the reviewers.

The paper can be improved in both breadth and depth. Specifically, `reviewer icES` asked for in-depth discussions about performance, security, and deployment aspects. `Reviewer YwtD` asked for an in-depth discussion regarding intra-process or intra-address space isolation, the type-1 and type-2 architectures, and the full-, para-, and hardware-assisted virtualization that are widely studied in the existing literature. `Reviewer jR75` asked for in-depth connections and relevance of virtualization categories, especially their intersections and modern use cases. Finally, `reviewer exqD` asked for discussions about use cases and implications.

In addition, thorough empirical evaluations or discussion of future research directions is essential for SoK papers, which are missing in the current version of the manuscript.

Although the manuscript in the current version cannot be accepted, we encourage the authors to revise their manuscript and consider a new submission here or elsewhere.

---

### Decision · Program_Chairs · 2022-11-14

**Decision:**

Reject

**Comment:**

Reviews will be available shortly